# AN AXIOMATIC APPROACH TO MODEL-AGNOSTIC CONCEPT EXPLANATIONS

## ABSTRACT

Concept explanation is a popular approach for examining how human-interpretable concepts impact the predictions of a model. However, most existing methods for concept explanations are tailored to specific models. To address this issue, this paper focuses on model-agnostic measures. Specifically, we propose an approach to concept explanations that satisfy three natural axioms: linearity, recursivity, and similarity. We then establish connections with previous concept explanation methods, offering insight into their varying semantic meanings. Experimentally, we demonstrate the utility of the new method by applying it in different scenarios: for model selection, optimizer selection, and model improvement using a kind of prompt editing for zero-shot vision language models.

## 1 INTRODUCTION

In today's world, as machine learning systems continue to have a growing impact on our lives, there is a pressing need to understand the reasoning behind their decisions. Unfortunately, the current black-box nature of ML models often leaves users puzzled. In response to this challenge, the field of explainable AI (XAI) has emerged, which aims to develop new types of explanations and methods. One type of explanation that stands out is *concept explanation*, which assesses the importance of a human-understandable concept to a prediction. For instance, in predicting the appearance of a "zebra" in an image, the presence of "stripes" is a human-understandable concept that may impact the prediction.

Several methods (Kim et al., 2018; Bai et al., 2022) have been developed to capture the importance of a concept in relation to a prediction. However, these methods are model-specific and require white-box access, which can be a limitation, for example, in cases where the model is proprietary and cannot be freely accessed for inspection. Moreover, there is often a lack of clarity surrounding the interpretation of these methods for concept explanation, the specific aspects they evaluate, and how they differ from each other.

To address these challenges, we aim to develop an axiomatic approach for determining the importance of a concept in relation to a prediction. We begin by establishing a few axioms, derive measures that are consistent with these axioms, and design an algorithm for estimating the measures. Additionally, we examine the connection between our method and previous research in the field.

The newly proposed framework offers a number of advantages:

1. **Axiomatically-based measures.** Our proposed measures are rooted in three axioms, with two of them having been implicitly assumed in prior studies. This axiomatic approach enables us to develop measures that are firmly established.

2. **Model-agnostic.** Our measures are model-agnostic, i.e., no need to know the internal workings of the model to apply the new measures.

3. **Integration of previous research.** We establish a connection between TCAV (Kim et al., 2018) (respectively completeness-aware (Yeh et al., 2020) explanations), and our approach.

4. **Semantic meaning of concept-explanation methods.** The new framework helps understand the semantic meaning of prior concept explanation methods: TCAV (Kim et al., 2018) is associated with necessity, while completeness-aware (Yeh et al., 2020) with sufficiency.

5. **Efficient Implementation.** We design an efficient implementation for estimating the new measures, which is more efficient than previous methods for concept-explanations.

We use the new method we developed for a variety of applications:

1. **Model selection.** Our approach is model-agnostic, allowing it to explain various models and thus facilitate model selection. We use it to make a comparison between logistic regression and random forest, assisting users in understanding different aspects of the learned models in order to enable them to make an informed decision as to which model to select.

2. **Optimizer selection.** We can use the new method to select between different optimizers. Namely, we apply our method on networks trained using different optimization algorithms: SGD and AdamW. Our method offers insights into the differences between these two optimization techniques, thereby assisting in the selection process.

3. **Model improvement via prompt editing.** Through our approach, we are able to improve model performance by mitigating the influence of irrelevant concepts. To elaborate, we use our method to detect concepts impacting the prediction but lack relevance to the classification task. We then identify them in the CLIP embedding space. We demonstrate that by editing the CLIP textual embedding and decreasing the representation of these irrelevant concepts, we can significantly enhance model performance.

Our method relies on having examples annotated with relevant concepts. However, when these annotations are unavailable, we show how to leverage recent developments in image captioning to automatically label concepts in a semantically meaningful manner.

## 2 RELATED WORK

**Post hoc explanations.** Post hoc explanations can offer valuable insights into how a model works without compromising the model's accuracy. These explanations provide additional information, such as feature-attributions (Lundberg & Lee, 2017; Ribeiro et al., 2016; 2018) that evaluate the contribution of each feature to the prediction. In addition to feature-attribution explanations, there are also example-based approaches like counterfactual explanations and influential examples. Counterfactual explanations (Wachter et al., 2017; Karimi et al., 2020) highlight what changes to input would have to be made to change the model's output. Influential examples (Koh & Liang, 2017; Koh et al., 2019) reveal which examples are the most important for a model's decision-making process. Overall, these post hoc explanations can help provide a better understanding of a model's inner workings. Concept explanations, the focus of this paper, are a form of post hoc explanations that focus on determining the impact of concepts on a model's predictions.

**Concept-based explanations.** Different studies have proposed methods to return the influence of concepts for neural network models (Kim et al., 2018; Bai et al., 2022; Yeh et al., 2020), unlike our approach these methods are not model-agnostic. Another line of research (Ghorbani et al., 2019; Yeh et al., 2020) has sought to learn relevant concepts for a given task. This paper proposes a novel axiomatic approach to post hoc concept explanations, which is model-agnostic and thus applicable to a wide range of models. In contrast to post hoc explanations, Koh et al. (2020); Espinosa Zarlenga et al. (2022); Yuksekgonul et al. (2023) propose a different approach to understanding concepts, using concept bottleneck models, where concepts are part of the model's hidden representation.

**Model-agnostic explanations.** Model-agnostic explanations are methods that produce explanations that do not require any knowledge about the internal workings of the model. The lack of access to the inner workings of a model can be particularly beneficial when the model is confidential and cannot be revealed. Numerous model-agnostic techniques are available for feature attribution (e.g., Lundberg & Lee (2017); Ribeiro et al. (2016)). However, when it comes to concept explanations, there are fewer model-agnostic approaches. Our approach is model-agnostic, which sets our approach apart from other methods such as those discussed in Kim et al. (2018); Bai et al. (2022) which require access to the model's gradients or hidden layers.

**Axiomatic approach to feature attribution explanations.** Feature attribution explanation assigns a value to each feature to represent its impact on the prediction. This explanation is a type

of concept explanation where the concepts are restricted to the feature values (for neural networks, the features are usually the internal representations of the inputs). To ensure a consistent and reliable explanation method, many researchers have suggested using an axiomatic approach to this type of explanation (Lundberg & Lee, 2017; Sundararajan et al., 2017). For example, the implementation invariance axiom proposed by Sundararajan et al. (2017) is model-agnostic and requires identical attributions for two functionally equivalent models.

**Understanding the explanations.** Concept explanations generalize the notion of feature attribution, which solely assigns significance to features in the prediction. However, even within this more narrow explanation category, different attribution methods (Lundberg & Lee, 2017; Ribeiro et al., 2016) produce varying and sometimes inconsistent results Krishna et al. (2022). While one approach to understand these differences is to show they have a similar mathematical structure (Han et al., 2022), a different approach is suggested in this paper. Specifically, we propose to understand the semantic differences when comparing different concept-explanations methods. We demonstrate that TCAV (Kim et al., 2018) is associated with the necessity of the concept to the prediction, while completeness-aware (Yeh et al., 2020) is associated with the sufficiency.

## 3 THE AXIOMATIC APPROACH

**Notation.** We use $\mathcal{X}$ to represent the set of input examples. The model to be explained is denoted by $h : \mathcal{X} \to \{-1, +1\}$, which takes inputs from $\mathcal{X}$ and outputs either $+1$ or $-1$ to indicate whether the prediction is true or false, respectively. The concept we are interested in measuring is denoted by $c : \mathcal{X} \to [-1, +1]$ (or $c : \mathcal{X} \to \{-1, +1\}$ in the discrete case), which maps inputs from $\mathcal{X}$ to values between $-1$ and $+1$, where $c(x) = 1$ indicates that the concept holds and $c(x) = -1$ indicates that it does not. While for simplicity we introduce the framework with the binary classification problem, it is straightforward to extend to the $k$-class case by treating it as $k$ one-vs-all classification problems. The probability distribution over examples is denoted by $p : \mathcal{X} \to [0, 1]$, which maps inputs from $\mathcal{X}$ to a probability, $\sum_x p(x) = 1$. The proofs are deferred to the appendix.

**Problem formulation.** Our goal is to identify a measure, denoted as $M$, which quantifies the influence of a concept $c$ on the predictions of an opaque model $h$; for example, consider $h$ as a predictor of the "zebra" class and $c$ is the concept of "stripes". This allows one to gain a deeper understanding of the model $h$ by making use of a concept that is easily understandable to humans. Towards achieving this goal, the first step is to examine the inputs of $M$. Specifically, the measure $M$ is a function maps the following parameters to $\mathbb{R}$: (i) the model $h$, (ii) the concept $c$, and (iii) the a priori probability $p(\cdot)$ over the examples.

### 3.1 AXIOM 1: LINEARITY WITH RESPECT TO EXAMPLES

The first axiom states that when measuring the impact of a concept, the impact on each example, which is determined solely by the example's properties, should be summed. For instance, when evaluating the impact of the concept "stripes" on predicting a "zebra", one should separately assess how impactful "stripes" are for each example $x$ and then sum everything together. The knowledge of how much stripes are impacting another example $x'$ does not influence how stripes are impacting example $x$. Mathematically, this axiom implies that $M$ can be expressed as

$$M(h, c, p) = \sum_{x \in \mathcal{X}} m(h(x), c(x), p(x)),$$

for some local impact function $m : \{-1, +1\} \times [-1, 1] \times [0, 1] \to \mathbb{R}$.

The linearity axiom is presumed in numerous other situations, such as the standard evaluation of a hypothesis using an *additive* loss function (e.g., Chapter 2 in (Shalev-Shwartz & Ben-David, 2014)). Additionally, previous techniques for concept explanations, such as TCAV, also made this assumption. For further information and comparison between various measures for concept explanations and our work, see Section 4.

## 3.2 AXIOM 2: RECURSIVITY

The second axiom focuses on the local measure $m$ and investigates how it relates the probability example, $p(x)$, to the properties of example $h(x)$ and $c(x)$. The axiom specifically aims to investigate how $m$ changes with respect to $p(x)$. This axiom draws parallels to the entropy recursivity axiom mentioned in (Csiszár, 2008).

The recursivity axiom requires that when splitting the probability $p(x)$ of an example between two new examples with the same $h(x)$ and $c(x)$ values, the value of $M$ remains unchanged. In other words, the axiom requires that replacing an example $x$ with two examples that have probabilities $p_1$ and $p_2$ such that $p_1 + p_2 = p(x)$ and the same $h(x)$ and $c(x)$ values, keeping all other examples unchanged, does not alter the value of $M$. We can write $M$ in the following way.

**Claim 3.1.** If $M$ satisfies the first two axioms and the range of $p$ is the rationals, then $M$ can be written as

$$M(h, c, p) = \sum_x p(x) \cdot s(h(x), c(x)),$$

for some function $s : \{-1, +1\} \times [-1, +1] \to \mathbb{R}$.

This axiom guarantees linearity in probability. Similar to the earlier axiom, prior methods for assessing concept explanations, including TCAV, have also implicitly relied on this assumption.

## 3.3 AXIOM 3: SIMILARITY

In the third axiom, we examine the term $s(h(x), c(x))$, which focuses on a single example with a certain probability (the exact probability is irrelevant for $s$). It aims to determine the contribution of $h(x)$ and $c(x)$ to the overall value of $M$. The similarity axiom asserts that for every example $x$, when $c(x)$ and $h(x)$ are close, $M$ will be high, and when they are far, $M$ will be low. To satisfy this axiom, any similarity function can be utilized. One natural choice is to take $s(h(x), c(x)) = c(x) \cdot h(x)$, which treat the class and concept similarly. As an illustration, consider evaluating the influence of "stripes" on a "zebra" predictor. When stripes are prominently featured in an instance (i.e., a high value of $c(x)$), it can influence the predictor only if a zebra is also present (i.e., $h(x) = 1$).

## 3.4 THE NEW MEASURES AND ESTIMATION ALGORITHM

We can explore different instantiations of $s$, either focus on a specific class or concept, or treat them symmetrically, resulting in the following measures.

**Definition 3.2.** Fix predictor $h : \mathcal{X} \to \{-1, +1\}$, concept $c : \mathcal{X} \to [-1, +1]$, and probability $p$ over the examples,

- (symmetric measure) For $s(h(x), c(x)) = c(x) \cdot h(x)$,

$$M(h, c, p) = \mathbb{E}_{x \sim p}[h(x)c(x)].$$

- (class-conditioned measure) For $s(h(x), c(x)) = c(x)$ if $h(x) = 1$ and otherwise undefined, after normalization

$$M(h, c, p) = \mathbb{E}_{x \sim p}[c(x)|h(x) = 1].$$

- (concept-conditioned measure) For $s(h(x), c(x)) = h(x)$ if $c(x)$ is high (i.e., $c(x) \geq \theta$ in the continuous case and $c(x) = 1$ in the discrete case) and otherwise undefined, after normalization

$$M_\theta(h, c, p) = \mathbb{E}_{x \sim p}[h(x)|c(x) \geq \theta].$$

Intuitively, class-conditioned measure quantifies how often the concept $c$ occurs in the class, e.g., if $c$ is always present when $h(x) = 1$, then the value of $M$ will be high; and concept-conditioned measure quantifies the probability of the class when the concept holds (i.e., exceeding the threshold), e.g., if whenever $c$ is present enough (i.e., $c(x) \geq \theta$), the class is 1, then $M$ will be high. Putting differently,

*class-conditioned measures* necessity *of c to h and concept-conditioned the* sufficiency

We can estimate the symmetric measure using the following simple algorithm (and in a similar manner for all the three measures).

---

**Algorithm 1** Algorithm for estimating the symmetric measure

---

1: **Input:** $n$ examples $x_1, x_2, \ldots, x_n$ sampled from distribution $p$ and labeled by $h$ and $c$
2: **return** $\frac{1}{n} \sum_{i=1}^{n} h(x_i)c(x_i)$

---

# 4 TCAV AND COMPLETENESS-AWARE EXPLANATIONS WITHIN THE AXIOMATIC FRAMEWORK

This section provides formal proof that previous research conducted on concept explanations can be viewed as a part of our new axiomatic approach, given certain assumptions. Specifically, we focus on two widely used methods: TCAV (Kim et al., 2018) and Completeness-Aware (Yeh et al., 2020) Explanations. A benefit of our approach is that it enables a faster implementation of these previous methods, as compared to their original implementation which required learning a model.

**Setting.** In prior research, ground truth labeling $\{(x_i, y_i)\}_i$ was utilized to assess the quality of concept explanations. However, as our objective is to evaluate the influence of explanations on the model, it is necessary for the evaluation measure to be independent of ground truth. For instance, when examining the influence of a particular profession on a loan decision, the actual loan repayment outcome becomes insignificant in understanding the impact of the profession *on the model*. Therefore, when comparing to previous research, we define $y_i = h(x_i)$ in order to eliminate the dependence on the ground truth.

## 4.1 COMPLETENESS-AWARE CONCEPT-BASED EXPLANATION

Yeh et al. (2020) propose a completeness score to evaluate the quality of a set of concepts for a prediction task. The primary term, i.e., the only one that depends on the concept, in the completeness score is the following (see Appendix B for the full completeness score),

$$S_{CACBE} \triangleq \sup_{g} \Pr_{x,y}[y = \arg\max_{y'} h_{y'}(g(c(x)))], \tag{1}$$

where $h_{y'}$ is the predicted probability of label $y'$, and $g$ is a function that maps the set of concepts to the feature space. In our notation, $\arg\max_{y'} h_{y'}$ is simply $h$. The score evaluates the quality of the explanation by analyzing the degree to which it is sufficient for decoding $h(x)$ with the help of $g$.

We show that the completeness score can be written as a simple aggregation of the concept-conditioned measure for the different classes, when $\mathcal{C}(x) = c(x)$, and the concepts and predictions are binary. Namely,

**Theorem 4.1.** For concept $c : \mathcal{X} \to \{-1, +1\}$ and predictor $h : \mathcal{X} \to \{-1, +1\}$, the following holds

$$S_{CACBE} = \frac{1}{2} + \frac{1}{2} \sum_{y=1,-1} |\mathbb{E}_x[h(x)|c(x) = y]| \cdot \Pr(c(x) = y). \tag{2}$$

In other words, Yeh et al. (2020) implicity applies Axiom 3 by treating each class independently and summing them all together to create the final similarity measure.

**Efficient implementation.** In order to compute the completeness score, one must first learn $g$, which can be a time-intensive process. In contrast, our approach eliminates the need for this learning phase. As a result, our measures can be computed more efficiently than those in the work of Yeh et al. (2020).

## 4.2 TESTING WITH CONCEPT ACTIVATION VECTORS (TCAV)

The problem of evaluating the impact of concepts was first introduced in the work of Kim et al. (2018). The approach involves using (i) a set of examples that are labeled with the concept of interest, and (ii) the gradient of the model's prediction for each example. They assumed the concept is

linearly separable, by a unit vector $v$, in the representation domain of the examples. Mathematically, Kim et al. (2018) quantifies the concept $c$ for example $x$ as $c(x) = g(x) \cdot v$, for some embedding function $g(\cdot)$. To ensure that $c(x) \in [-1, +1]$, we make the assumption that unit norm embeddings are used. As a side-note, subsequent research (Bai et al., 2022) relaxed the linearity assumption and employed the concept's gradient, rather than utilizing $v$.

The TCAV definition involves computing the ratio of examples in a specific class, $X_k$, that has a positive score. The score is computed as the dot product between the gradient of the model's prediction with respect to the representation $g(x)$ and the concept unit vector $v$. Quantitatively, the TCAV definition is

$$\text{TCAV} := \frac{\{|x \in X_k : S(x) > 0\}|}{|X_k|},$$

where $S(x) = \nabla f(g(x)) \cdot v$ and $f$ is some function. Our focus is on a continuous version of TCAV: $\text{TCAV}_{con} := \frac{\sum_{x \in X_k} S(x)}{|X_k|}$ The work of Kim et al. (2018) investigates the impact of a concept on various layers of a neural network by utilizing the representations in those layers as $g(x)$ and the corresponding subsequent layers as $f(\cdot)$. In this work we treat the last layer as $g(\cdot)$, i.e., we focus on models $h(x) \triangleq sign(f(g(x))) = sign(w_h \cdot g(x) - \theta_h)$, where $w_h$ is a unit norm. Thus, we can infer that $\nabla f(g(x)) = w_h$ and $S(x) = w_h \cdot v$. Our next theorem proves that $\text{TCAV}_{con}$ is similar to class-conditioned measure.

**Theorem 4.2.** Fix a concept $c(x) = g(x) \cdot v$ for some $v$ and $g$, and a predictor $h(x) = sign(w_h \cdot g(x) - \theta_h)$. For any $\epsilon, \delta \in (0, 1)$, if there are at least $2 \ln(1/\delta)/\epsilon^2$ concept-labeled examples to calculate TCAV', and $\theta_h \geq 1 - \epsilon^2/8$, then with probability at least $1 - \delta$

$$|\mathbb{E}_{x \sim p}[c(x)|h(x)] - \text{TCAV}_{con}| < \epsilon.$$

The theorem teaches us that TCAV measures the necessity of a concept for a prediction, which differs from the approach in the work of Yeh et al. (2020) that measures the sufficiency of the concept.

**Efficient implementation.** Both TCAV and its follow-up work (Bai et al., 2022) achieve their objective by learning the concept. TCAV learns the vector $v$ and the follow-up work learns the concept to find its gradient. In contrast to these techniques, our approach does not require learning the concept. As a result, calculating our measures is more efficient than TCAV and its follow-up work (Kim et al., 2018; Bai et al., 2022).

## 5 EXPERIMENTS

In this section, we provide several experiments to demonstrate the validity and applicability of our approach. In Section 5.1 we validate that our measures capture the semantic relationship between the predictor and the concept. In Section 5.2 we explore different applications of our method. Finally, in Section 5.3 we show that concepts can be automatically labeled by an image captioner.

### 5.1 NECESSARY AND SUFFICIENT EXPLANATIONS

As a proof of concept, this section validates the semantic meaning of the measures through two scenarios. In the first scenario $h(x)$ predicts the fine-grained Felidae classes (e.g., Persian cat or cheetah). When $c(x)$ is the "Felidae" concept, the concept is necessary, and we expect the class-conditioned measure to be high. When $c(x)$ is the irrelevant "wolf" concept we expect the measure to be low. In the second scenario, when $h(x)$ predicts "Felidae" and $c(x)$ is the fine-grained Felidae concepts, the concept is sufficient for the predictor, and thus we expect the concept-conditioned measure to be high. If $h(x)$ predicts "wolf", the concepts are irrelevant and we expect the measure to be low.

Empirically, we show in Figure 1 that a pretrained ResNet-18 (He et al., 2016) on ImageNet (Deng et al., 2009) exhibits this expected behavior. Specifically, among the 1000 classes in ImageNet, we consider the classes under the "Felidae" hierarchy, these include class ids ranging from 281 to 293. When predicting the more coarse class (i.e. when we try to predict whether the image contains a Felidae or not, rather than recognizing the fine-grained type of Felidae), for simplicity we set $h(x) = 1$ whenever the ResNet outputs any index in $\{281, \ldots, 293\}$.

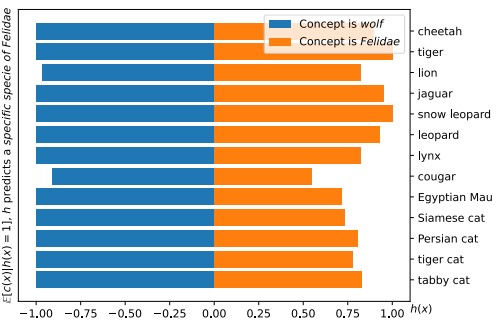 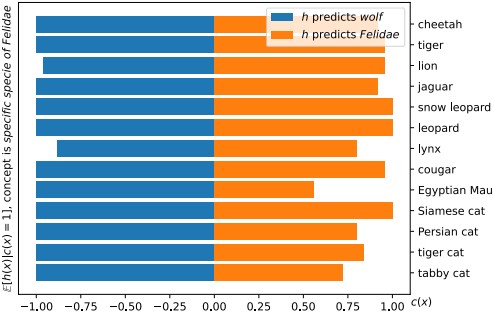

Figure 1: Necessary and sufficient concepts under our proposed measures. **Left**: $\mathbb{E}[c(x)|h(x) = 1]$, where $h(x)$ predicts the fine-grained Felidae classes (e.g. Persian cat or cheetah), $c(x)$ is the "Felidae" or "wolf" concept. **Right**: $\mathbb{E}[h(x)|c(x) = 1]$ where $h(x) = 1$ means the image is a wolf or Felidae, $c(x)$ is the fine-grained Felidae concepts.

## 5.2 APPLICATIONS

In this section, we explore three different experiments demonstrating the applicability of the newly devised measures. We conduct experiments on the a-Pascal dataset (Farhadi et al., 2009). This dataset contains 6340 training images and 6355 testing images, distributed unevenly over 20 classes. Each image is associated with a concept vector in $\{0, 1\}^{64}$, with concepts like "wing" and "shiny". The true concepts are labeled by human experts, and in Section 5.3 we show that current image captioners can achieve reliable performance in labeling concepts as well. Unless otherwise specified, we use the human-labeled concepts in the experiments.

### 5.2.1 LOGISTIC REGRESSION VERSUS RANDOM FOREST EXPLANATIONS

A key advantage of our approach is its ability to operate with any model, allowing us to apply our measures to both logistic regression (LR) and random forest (RF) models. This sets our approach apart from previous research that primarily focused on neural networks. In this section, we utilize these measures to compare LR and RF models.

We trained the LR classifier with a $\ell_2$ regularizer. The RF classifier has 100 trees and each has a maximum depth of 15. Both models are trained using Pedregosa et al. (2011), hyperparameters are set to default unless stated otherwise. Both models achieve 87% accuracy. We plot the model explanation with $\mathbb{E}[h(x)|c(x) = 1]$ and $\mathbb{E}[c(x)|h(x) = 1]$, see Figure 2. As a comparison, we also draw $\mathbb{E}[c(x)|y = 1]$ and $\mathbb{E}[y|c(x) = 1]$, labeled as "ground truth" in the figures.

Our approach provides valuable insights into the dissimilarities of the models, despite LR and RF achieving almost identical accuracy. Specifically, when predicting the cat class, RF weights more on the concepts "tail", "torso", and "leg", while LR puts more weight on "eye". When predicting the chair class, RF is more sensitive to the concept "leather" compared to LR, which is not aligned with the true data distribution. These insights can be valuable in various scenarios, such as identifying spurious correlations and providing guidance during model selection. For instance, if a concept exhibits a high measure but lacks relevance, it can indicate the presence of spurious correlations. When faced with the choice between several models of equal quality, opting for the one that utilizes a greater number of relevant concepts can be a wise choice.

### 5.2.2 SGD VERSUS ADAMW

Our proposed method also provides a convenient way to study the behavior of different optimizers on the semantic level. The conventional wisdom has been that SGD finds a minimum that generalizes better while ADAM finds a local minimum faster (Zhou et al., 2020), we are able to demonstrate what semantic concepts in the images are picked up by each optimizer, which can potentially lead to an explanation to the discrepancies in the optimizers' behaviors in addition to generalization ability.

We use ResNet-18 pretrained on ImageNet, and finetune with the following optimizers (unspecified hyperparameters are set to PyTorch defaults): (a) SGD, last linear layer learning rate $1e - 3$, other

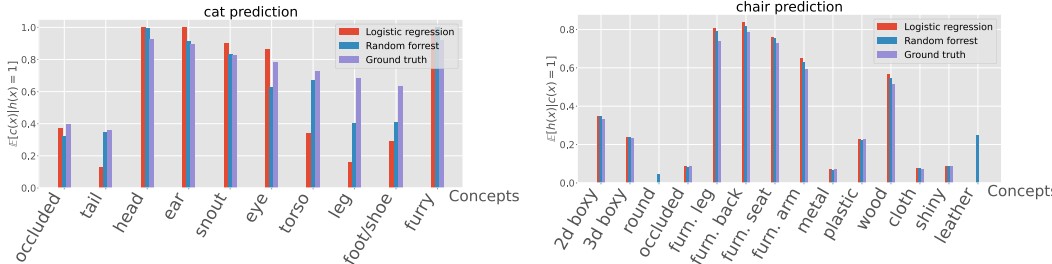

Figure 2: Logistic regression versus random forest. In both figures, the $x$-axis represents the concepts with positive measures, and the $y$-axis corresponds to a specific measure at interest. **Left**: $\mathbb{E}[c(x)|h(x) = 1]$, where $h(x) = 1$ predicts class "cat". **Right**: $\mathbb{E}[h(x)|c(x) = 1]$, where $h(x) = 1$ predicts class "chair".

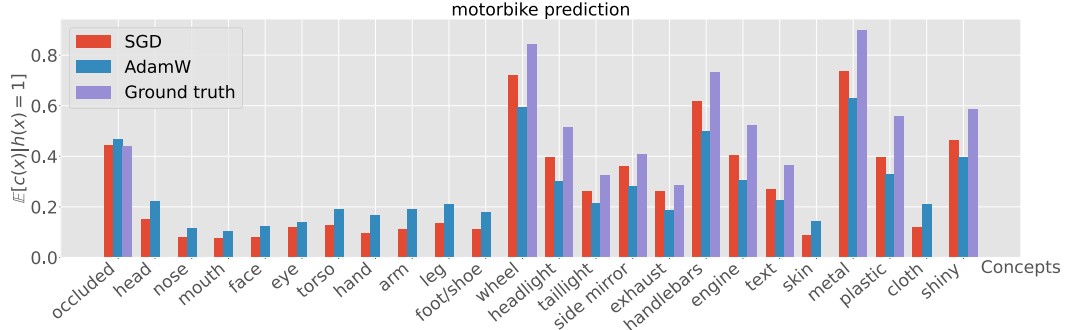

Figure 3: SGD versus AdamW with $\mathbb{E}[c(x)|h(x) = 1]$, where $h(x)$ predicts "motorbike" class.

layer learning rate $1e-9$, momentum $0.9$, weight decay $5e-4$. (b) AdamW, last linear layer learning rate $1e-3$, other layer learning rate $1e-9$. Both models are trained $100$ epochs, the SGD model achieves a test accuracy of $58.46\%$ and the AdamW model achieves $60.26\%$. Based on Figure 3, AdamW exhibits a tendency to overemphasize the motorbike class by giving excessive importance to irrelevant features such as "leg," while assigning insufficient weight to relevant features like "engine." This is in contrast to both SGD and the ground truth. This observation suggests that SGD is more suitable optimizer for this task.

### 5.2.3 PROMPT EDITING

This section demonstrates how our new approach can enhance model performance through prompt editing. The recent development of vision-language models like CLIP (Radford et al., 2021) enables exceptional zero-shot classification performance via prompt engineering. These models have an image encoder $f(x) : \mathbb{R}^d \to \mathbb{R}^h$ and a text encoder $g(t) : \mathbb{R}^p \to \mathbb{R}^h$. These encoders are trained contrastively to align the matching $f(x)$ and $g(t)$, where the text $t$ is the true caption of the image $x$. During inference time, given a set of $\mathcal{Z}$ classes, for each $z \in \mathcal{Z}$ we can construct a class prompt "a photo of $\{t_z\}$", and for an input image $x$, we predict: $\hat{z} = \arg\max_{z \in \mathcal{Z}} f(x) \cdot g(t_z)$. We use the CLIP model with ViT-B16 (Dosovitskiy et al., 2020) vision backbone on the a-Pascal dataset in this experiment, achieving a zero-shot test accuracy of $78.2\%$ and an F1-score of $0.6796$. However, we identify 10 class prompts that mistakenly excessive emphasize on 13 irrelevant concepts. For example, in Figure 4, when conditioning on the original CLIP prompts to predict "bottle", there are many predicted images containing irrelevant concepts like "head", "ear", and "torso".

To counter the false emphasize, we edit the original class prompts of these 10 classes by subtracting the irrelevant concepts. In particular, considering an irrelevant concept $w$ for class $z$, we use $g(t_z) - \lambda \cdot g(w)$ as the new prompt embedding for classifying class $z$, where $\lambda < 1$ is a scalar that prevents us from over-subtracting. Heuristically we pick $\lambda = 0.1$ based on trial-and-error using the training data, however, one can certainly learn $\lambda$ in a few-shot fashion. See more details in the appendix.

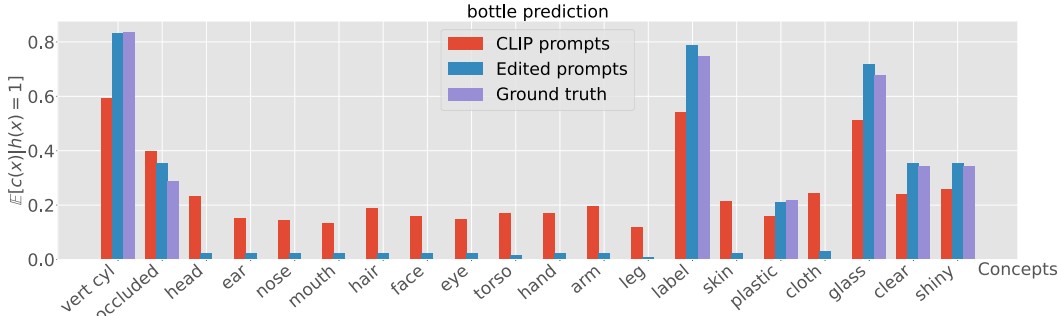

Figure 4: Prompt editing with $\mathbb{E}[c(x)|h(x) = 1]$ where $h(x) = 1$ predicts class "bottle".

The edited prompts improve the F1-score to $0.7042$. Moreover, we can clearly see that the edited prompts align more with human intuition. Figure 4 shows that irrelevant concepts have a smaller impact on edited prompts' predictions than on CLIP's predictions.

## 5.3 Automatic Concept Labeling with Image Captioner

Here we show that the state-of-the-art image captioners can reliably label the concepts in images. We use BLIP-2 (Li et al., 2023) with the pretrained FlanT5XL backbone (Chung et al., 2022) with the prompt "`Question: Does the {classname} in the picture have {concept}? Short Answer:`". The maximum generation length is set to 1. This configuration almost always only returns either "yes" or "no". To reduce the variance in answers, we ask ChatGPT for 10 synonyms to the above-mentioned prompt. In total, we gather 11 questions for each of the 64 concepts. The final decisions are made based on thresholding the number of "yes" answers. Following Bai et al. (2022), we focus on the accuracy and recall metrics for concept retrieval. Recall@k means if the number of "yes" exceeds $k$, we label the concept as positive and compute the recall correspondingly. The same applies to acc@k. We try several different values of the thresholds and present the result in Table 1.

Table 1: Concept labeling accuracy and recall at different thresholds, the model is FlanT5XL.

| Acc@1 | Acc@3 | Acc@5 | Recall@1 | Recall@3 | Recall@5 |
|---|---|---|---|---|---|
| 0.5120 | 0.7366 | 0.8071 | 0.9483 | 0.8520 | 0.7672 |

## 6 Conclusion

In the paper, we proposed a novel axiomatic approach for concept explanations and developed an efficient algorithm to estimate it. Moreover, we investigated the connection between our approach and earlier techniques for concept explanation. Subsequently, we harnessed the power of our newly developed algorithm across a range of scenarios, showcasing its many advantages: it provides guidance to model selection, aids in choosing the optimizer for training neural networks, and, through prompt editing, enhances models' performance.

There are several directions for further research. Firstly, we suggest investigating concept learning and discovery to expand our understanding of how concepts are represented and identified. Given that there are potentially infinite concepts, efficiently learning a concise set of relevant concepts is essential for ensuring interpretability without human intervention. Secondly, it would be interesting to explore the semantic meaning of different similarity functions to better understand how they influence concept explanations. Finally, we propose examining the use of concept explanation for transfer learning, where the method may help improve generalization performance and reduce overfitting.

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

## A   PROOFS FOR SECTION 3

**Claim 3.1.** If $M$ satisfies the first two axioms and the range of $p$ is the rationals, then $M$ can be written as

$$M(h, c, p) = \sum_x p(x) \cdot s(h(x), c(x)),$$

for some function $s : \{-1, +1\} \times [-1, +1] \to \mathbb{R}$.

*Proof.* From the first axiom we know that $M$ can be written as

$$M(h, c, p) = \sum_x m(h(x), c(x), p(x)),$$

for some $m$. For any example $x_0$, the recursivity axiom tells us that

$$m(h(x_0), c(x_0), p_1 + p_2) + \sum_{x \neq x_0} m(h(x), c(x), p(x))$$

is equal to

$$m(h(x_0), c(x_0), p_1) + m(h(x_0), c(x_0), p_2) + \sum_{x \neq x_0} m(h(x), c(x), p(x))$$

So overall, when focusing on $m$ as a function of $p$, the recursivity axiom implies that

$$m(\cdot, \cdot, p_1 + p_2) = m(\cdot, \cdot, p_1) + m(\cdot, \cdot, p_2) \tag{3}$$

It is a well-known fact that the last equation implies that $m$ is linear with respect to the probability $p(x)$ over the rational numbers. For the sake of completeness, this fact is proved in the next claim. Thus, $m$ can be written as

$$m(p(x), c(x), h(x)) = p(x) \cdot s(c(x), h(x)).$$

$\square$

**Claim A.1.** $m$ is linear with respect to $p$.

*Proof.* Recall that a linear function f(x) is a function that satisfies two properties: (1) additivity: $f(x + y) = f(x) + f(y)$ and (2) $f(\alpha x) = \alpha f(x)$ for all $\alpha$. The first property is Equation 3. Thus, we need to prove that this equation also imply property (2), when $\alpha$ is rational. We will show it in three steps, first when $\alpha$ is integer, then when $1/\alpha$ is integer, and finally when $\alpha$ is rational, i.e., $\alpha = a/b$ when $a$ and $b$ are integers.

For integer $a$, and probability $p$, it holds that

$$m(ap) = m(p + p + \ldots + p) = am(p).$$

For integer $1/a$, it holds that

$$m(p) = m(\frac{1}{a} \cdot ap) = \frac{1}{a} \cdot m(ap),$$

which implies that,

$$am(p) = m(ap).$$

For rational $a/b$ it holds that

$$m(a/b \cdot p) = m(a \cdot 1/b \cdot p) = am(1/b \cdot p) = \frac{a}{b} m(p).$$

$\square$

## B   PROOFS FOR SECTION 4

For a model $h$, the completeness score Yeh et al. (2020) is defined as

$$\frac{\sup_g \Pr_{x,y}[y = \arg\max_{y'} h_{y'}(g(\mathcal{C}(x))] - a_r}{\Pr_{x,y}[y = \arg\max_{y'} f_{y'}(x)] - a_r},$$

where $\mathcal{C}(x)$ are the concepts extracted from the input instance $x$, $a_r$ is the accuracy of a random prediction to equate the lower bound of completeness score to 0, and $f$ is the neural network to be explained. In our setting, the denominator is equal to 1. Thus, up to normalization, the completeness score is equal to

$$\sup_g \Pr_{x,y}[y = \arg\max_{y'} h_{y'}(g(\mathcal{C}(x)))].$$

In this paper we focus on measures of one concept, hence $\mathcal{C}(x) = c(x)$. And recall that we denoted

$$S_{CACBE} \triangleq \sup_g \Pr_{x,y}[y = \arg\max_{y'} h_{y'}(g(c(x)))], \tag{4}$$

Now we are ready to prove the Theorems in Section 4.

**Theorem 4.1.** For concept $c : \mathcal{X} \to \{-1, +1\}$ and predictor $h : \mathcal{X} \to \{-1, +1\}$, the following holds

$$S_{CACBE} = \frac{1}{2} + \frac{1}{2} \sum_{y=1,-1} |\mathbb{E}_x[h(x)|c(x) = y]| \cdot \Pr(c(x) = y). \tag{2}$$

*Proof.* Focus on the LHS of Equation (2). Partition all examples into four parts, depending on their prediction $h(x)$ and their concept $c(x)$. Let $g : \{-1, +1\} \to \mathcal{X}$, and recall that at the beginning of Section 4 we set $y \triangleq h(x)$. Then, the LHS is equal to

$$\Pr_x(1 = h(g(1))|c(x) = 1 \wedge h(x) = 1]) \Pr(c(x) = 1 \wedge h(x) = 1)$$
$$+ \Pr_x(-1 = h(g(1))|c(x) = 1 \wedge h(x) = -1]) \Pr(c(x) = 1 \wedge h(x) = -1)$$
$$+ \Pr_x(1 = h(g(-1))|c(x) = -1 \wedge h(x) = 1]) \Pr(c(x) = -1 \wedge h(x) = 1)$$
$$+ \Pr_x(-1 = h(g(-1))|c(x) = -1 \wedge h(x) = -1]) \Pr(c(x) = -1 \wedge h(x) = -1)$$

To maximize the first two terms, $h(g(1))$ should be 1 if

$$\Pr(c(x) = 1 \wedge h(x) = 1) > \Pr(c(x) = 1 \wedge h(x) = -1)$$

and otherwise $h(g(1)) = -1$. Similar argument applies to the last two terms. Hence, the LHS in Equation 2 is equal to

$$\begin{aligned} &\max(\Pr(c(x) = 1 \wedge h(x) = 1), \Pr(c(x) = 1 \wedge h(x) = -1)) \\ &+ \max(\Pr(c(x) = -1 \wedge h(x) = -1), \Pr(c(x) = -1 \wedge h(x) = -1)) \end{aligned} \tag{5}$$

We can rewrite the above probabilities, for $y = -1, +1$

$$\Pr(h(x) = +1 \wedge c(x) = y) = \mathbb{E}_x \left[ \frac{h(x) + 1}{2} \mid c(x) = y \right] \cdot \Pr(c(x) = y)$$
$$\Pr(h(x) = -1 \wedge c(x) = y) = \mathbb{E}_x \left[ \frac{-h(x) + 1}{2} \mid c(x) = y \right] \cdot \Pr(c(x) = y)$$

In each term of Equation 5, taking the max translates to taking an absolute value, as can be seen by the following equations

$$\max \left( \mathbb{E}_x \left[ \frac{h(x) + 1}{2} \mid c(x) = y \right] \cdot \Pr(c(x) = y), \mathbb{E}_x \left[ \frac{-h(x) + 1}{2} \mid c(x) = y \right] \cdot \Pr(c(x) = y) \right)$$
$$= \frac{1}{2} \left( \max(\mathbb{E}_x[h(x) \mid c(x) = y], -\mathbb{E}_x[h(x) \mid c(x) = y]) + 1 \right) \cdot \Pr(c(x) = y)$$
$$= \frac{1}{2} (|\mathbb{E}_x[h(x)|c(x) = y]| + 1) \cdot \Pr(c(x) = y),$$

which is equal to a single term in the sum on the RHS of Equation 2, thus proving the claim. $\qquad \square$

**Theorem 4.2.** Fix a concept $c(x) = g(x) \cdot v$ for some $v$ and $g$, and a predictor $h(x) = sign(w_h \cdot g(x) - \theta_h)$. For any $\epsilon, \delta \in (0, 1)$, if there are at least $2\ln(1/\delta)/\epsilon^2$ concept-labeled examples to calculate TCAV', and $\theta_h \geq 1 - \epsilon^2/8$, then with probability at least $1 - \delta$

$$|\mathbb{E}_{x \sim p}[c(x)|h(x)] - \mathrm{TCAV}_{con}| < \epsilon.$$

*Proof.* Before stating the full proof, here are the main ideas. The proof begins with estimating the class-conditioned measure using the formula $\frac{\sum_{x \in X_k} c(x)}{|X_k|}$, which is closely related to the TCAV definition. Then we need to show that $c(x)$ is similar to $S(x)$ when $x$ is labeled as 1. Such examples satisfy that $x$ and $w_h$ are close to each other. This intuitively holds because $c(x) = g(x) \cdot v$ measures the closeness of $x$ and $v$, while $S(x) = w_h \cdot v$ measures the closeness of $w_h$ and $v$. Since $x$ and $w_h$ are close, so are $S(x)$ and $c(x)$, which proves the theorem. To prove the theorem formally we apply CauchySchwarz inequality, triangle inequality, and Hoeffdings inequality.

Now we countinue with the formal proof. Since $v$ and all the embeddings are normalized to have a unit norm, the Cauchy-Schwarz inequality implies that the value of $c(x)$ falls within the range of $[-1, 1]$. Therefore, we can use Hoeffding's inequality as shown in Fact B.1, to prove that the average concept in the class is a reliable estimator,

$$\left| \mathbb{E}_{x \sim p}[c(x)|h(x)] - \frac{\sum_{x \in X_k} c(x)}{|X_k|} \right| < \frac{\epsilon}{2}. \tag{6}$$

Therefore, it is sufficient to provide a bound for the following term

$$\left| \frac{\sum_{x \in X_k} c(x)}{|X_k|} - \text{TCAV}_{con} \right| = \left| \frac{\sum_{x \in X_k} g(x) \cdot v}{|X_k|} - \frac{\sum_{x \in X_k} w_h \cdot v}{|X_k|} \right|.$$

By applying the triangle inequality, it is satisfactory to establish a bound for each of the inner terms in the sum in order to prove our claim. Specifically, we need to bound the following term to support our assertion.

$$|g(x) \cdot v - w_h \cdot v| = |(g(x) - w_h) \cdot v|$$

By CauchySchwarz inequality, the last term is bounded by

$$\|g(x) - w_h\| \cdot \|v\| = \|g(x) - w_h\| = \sqrt{\|g(x)\|^2 + \|w_h\|^2 - 2g(x) \cdot w_h} = \sqrt{2(1 - g(x) \cdot w_h)},$$

where the first equality follows from the fact that $v$ is a unit norm and the last equality follows from the fact that $g(x)$ and $w_h$ are unit norms. Knowing that $x$ belongs to $X_k$, we can deduce that $w_h \cdot g(x) - \theta_h$ is greater than zero. Implying that $g(x) \cdot w_h \geq 1 - \epsilon^2/8$, which is equivalent to $1 - g(x) \cdot w_h \leq \epsilon^2/8$. This proves that

$$|g(x) \cdot v - w_h \cdot v| \leq \frac{\epsilon}{2}.$$

The assertion's statement follows when combining both the last inequality and Inequality 6. $\square$

**Fact B.1** (Hoeffding's inequality). For $X_1, \ldots, X_n \in [-1, 1]$ independent random variables

$$\Pr\left( \left| \frac{1}{n} \sum_i \mathbb{E}[X_i] - \frac{1}{n} \sum_i X_i \right| > \epsilon \right) < \exp(-n\epsilon^2/2)$$

## C  EXPERIMENT DETAILS

### C.1  PROMPT EDITING WITH FEW-SHOT DATA

As mentioned in section 5, we can leverage few-shot data when editing prompts with the help of the new measures. We pick 10 classes of which the measure $\mathbb{E}[c(x)|h(x) = 1]$ does not align with the ground truth (which also aligns with human intuition). In particular, "bicycle", "bird", "bottle", "chair", "diningtable", "horse", "motorbike", "sofa", "train", "tvmonitor" are the selected classes. In Figure 5 we see $\mathbb{E}[c(x)|h(x) = 1]$ on all 20 classes, and among them the above-mentioned 10 classes are selected for prompt editing. For simplicity, we focus only on a subset of 13 concepts that we want to modify (specifically here, we only want to subtract them), which include "head", "ear", "nose", "mouth", "hair", "face", "eye", "torso", "hand", "arg", "leg", "foot/shoe", "skin".

To edit the prompt, for any $z$ in the above classes, we first get the normalized[1] text embedding $\mathcal{I}_z$ of its prompt "a photo of {z}". Then for each concept $c \in \mathcal{C}$ that we want to subtract, we get its normalized text embedding $\mathcal{I}_c$. Finally, we use $\mathcal{I}_z - \lambda \cdot \frac{1}{|\mathcal{C}|} \sum_{c \in \mathcal{C}} \mathcal{I}_c$ as the prompt embedding, where $\lambda = 0.1$. We see that after editing, the measure $\mathbb{E}[c(x)|h(x) = 1]$ aligns with the ground truth more, see Figure 5.

Instead of heuristically setting $\lambda = 0.1$ for every class, we can instead use $\mathcal{I}_z - \cdot \lambda_z \frac{1}{|\mathcal{C}|} \sum_{c \in \mathcal{C}} \mathcal{I}_c$, and learn $\lambda_z$. We use $k = 1, 2, 4, 8, 16$ shot per class for training. We train using AdamW with a weight decay of 1 for 1000 epochs. They achieve a superior F1-score compared to the original CLIP prediction (see Table 2), and also a superior accuracy (see Table 3) for large enough $k$.

---

[1] CLIP always normalizes its embedding to the unit norm.

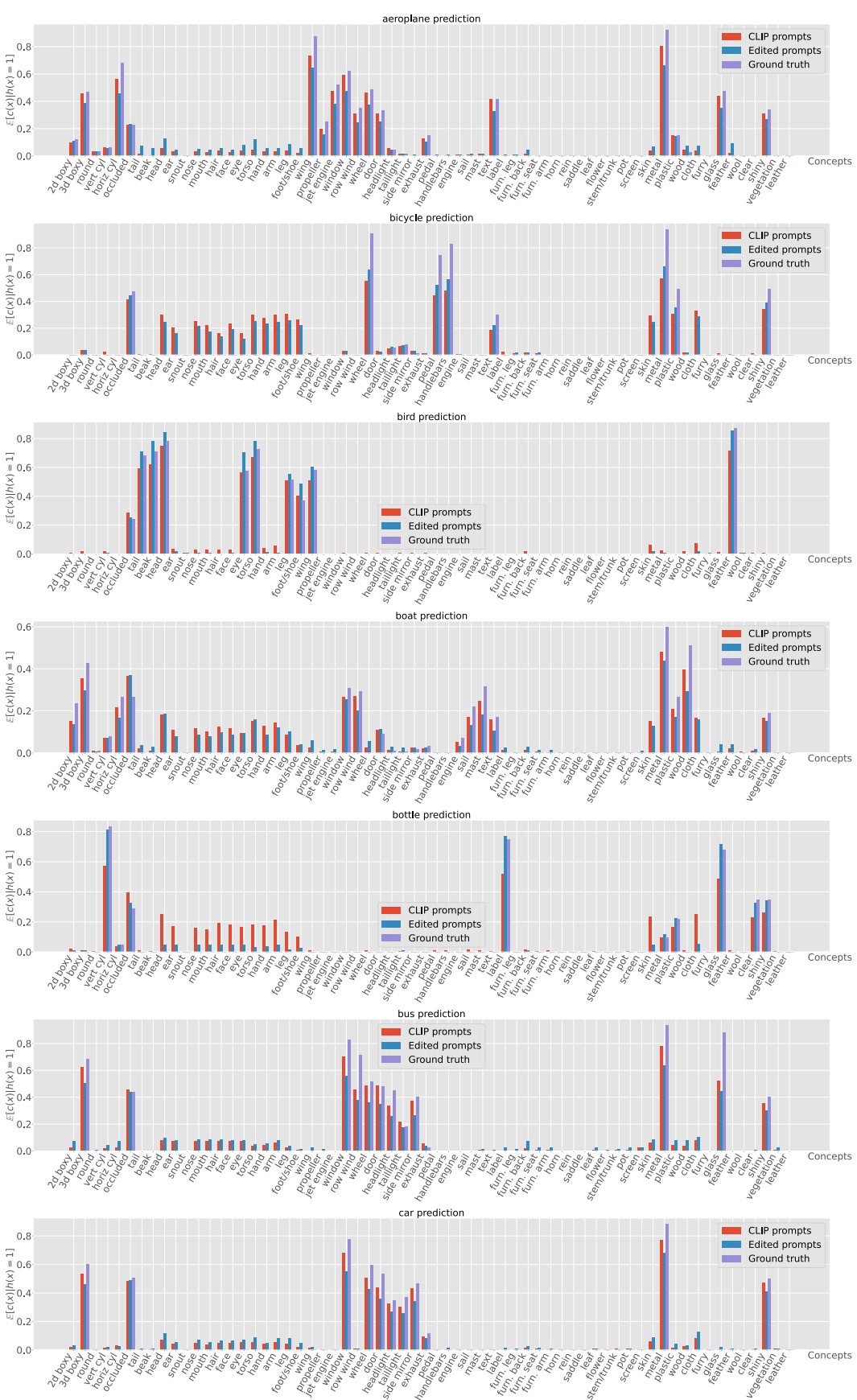

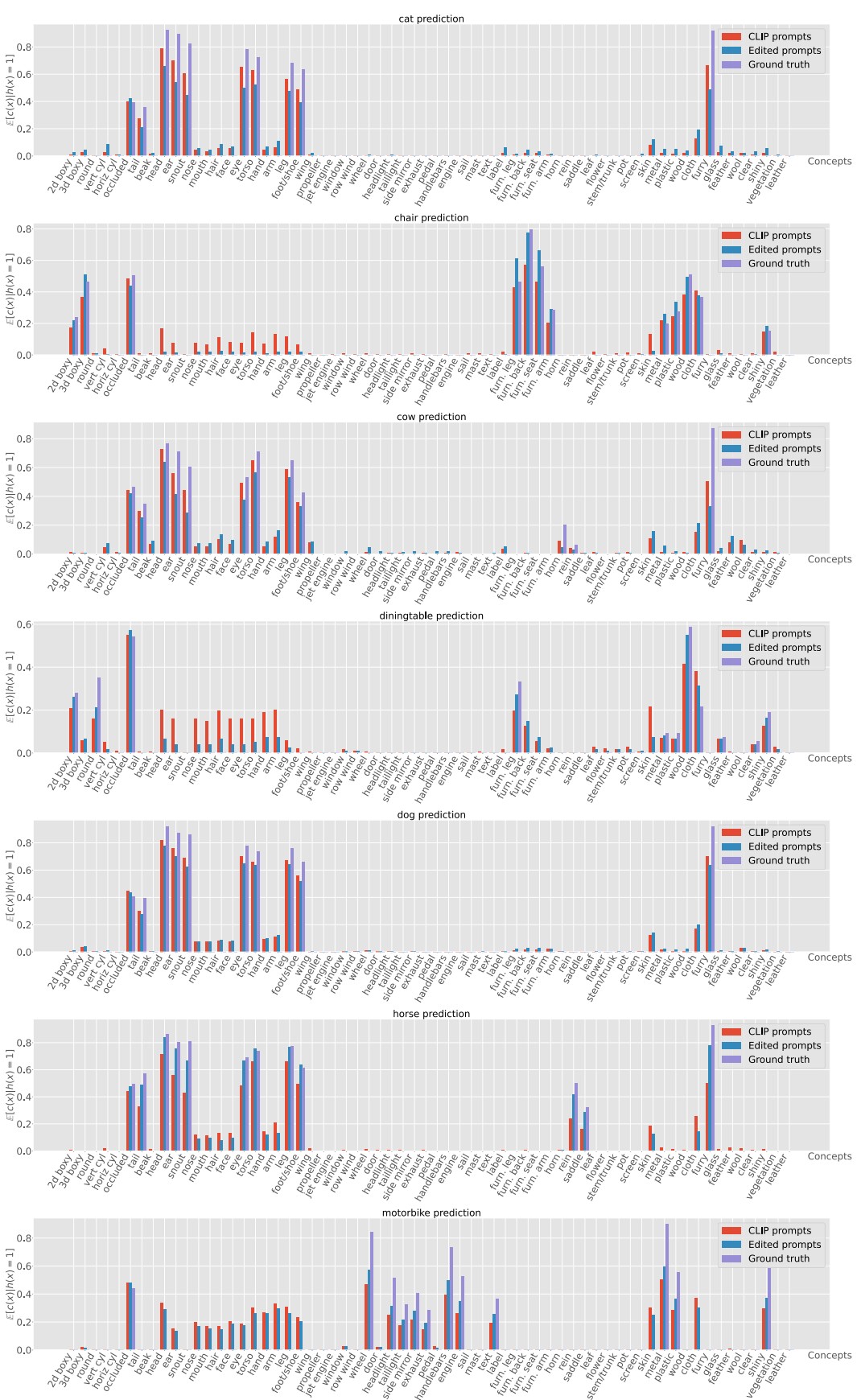

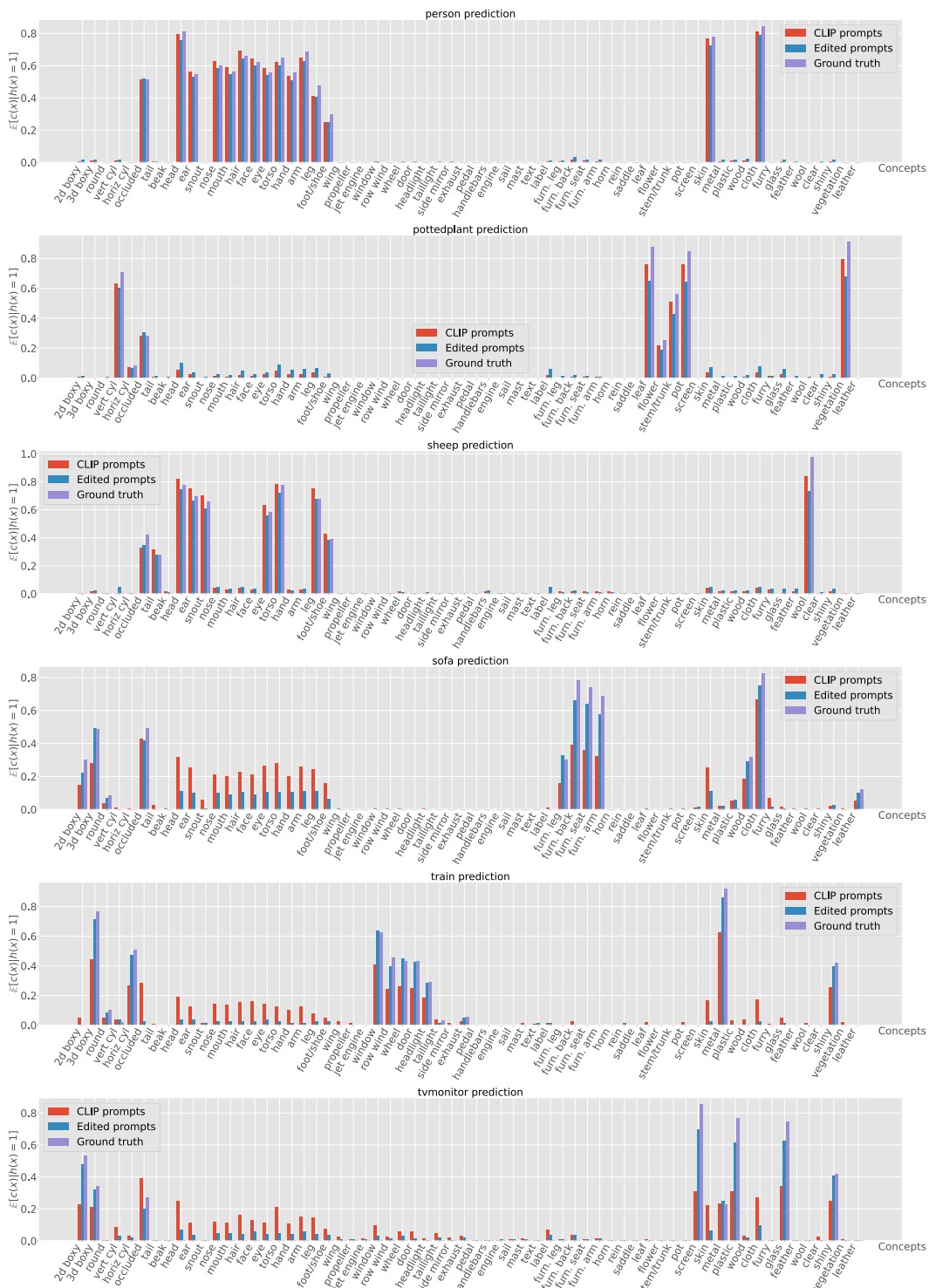

Figure 5: $\mathbb{E}[c(x)|h(x) = 1]$ with original CLIP prompts, edited prompts, and ground truth.

Table 2: F1-score (standard deviation) of the original CLIP prediction and few-shot learned $\lambda_z$

| Original | $k = 1$ | $k = 2$ | $k = 4$ | $k = 8$ | $k = 16$ |
|---|---|---|---|---|---|
| 0.6796 | 0.7001 (0.0073) | 0.6903 (0.0089) | 0.6996 (0.0037) | 0.6991 (0.0044) | 0.7003 (0.0027) |

Table 3: Accuracy (standard deviation) of the original CLIP prediction and few-shot learned $\lambda_z$

| Original | $k = 1$ | $k = 2$ | $k = 4$ | $k = 8$ | $k = 16$ |
|---|---|---|---|---|---|
| 0.7820 | 0.7637 (0.0138) | 0.7795 (0.0047) | 0.7762 (0.0047) | 0.7848 (0.0024) | 0.7844 (0.0025) |

