# OpenReview forum: "An Axiomatic Approach to Model-Agnostic Concept Explanations"
_ICLR.cc/2024/Conference — Submitted to ICLR 2024_

### Official Review · Reviewer_NFev · 2023-10-30

**Soundness:** 2 fair
**Presentation:** 2 fair
**Contribution:** 2 fair
**Rating:** 3
**Confidence:** 4

**Summary:**

In this paper, authors propose a method for concept explanations that satisfy three axioms. Such explanation is verified to provide insights into model selection, optimizer selection, and model improvement.

**Strengths:**

+ Concept explanation is an important direction in explainable AI research.
+ It is interesting to connect the proposed method with existing methods including TCAV.

**Weaknesses:**

- It seems that the proposed method is only applicable to datasets with pre-defined concept annotations c(x).
- Though authors show some connections between the proposed method and two existing methods, there still lacks baseline comparisons in the experiment part. For example, why not compare the proposed method with existing methods (such as [cite 1,2]) for model improvement.
- As typical concept explanations usually provide visualizations of extracted concepts. It will help a lot if authors also provide such visualizations, and compare their method with existing concept explanations from this perspective.


[cite 1] This Looks Like That: Deep Learning for Interpretable Image Recognition (NeurIPS 2019)

[cite 2] Concept-level Debugging of Part-Prototype Networks (ICLR 2023)

**Questions:**

none

---

> ### Author Response · Authors · 2023-11-21
> **Thanks for your review**
>
> We thank the review for their review, please see our responses below.
>
> - Note that the proposed method is not only applicable to datasets with pre-defined concept annotations. In Section 5.3 we discussed how to automatically label concepts.
>
>
> - Thank you for the references you provided, we will add them in future versions of the paper. However, please note that in this paper, we adopt a black-box approach and intentionally avoid limiting ourselves to neural networks. In the related work section, specifically at the end of the paragraph discussing "concept-based explanations", we discussed papers where concepts constitute part of the model’s hidden representation, similar to [cite 1] and [cite 2].
>
> - We appreciate the reviewer's perspective on the value of visualizations in evaluating XAI methods and understand their importance in certain contexts. However, we hold a different view in this instance. We believe that our approach, which focuses on more quantifiable and objective metrics, offers a complementary perspective to the prevalent use of visualizations. This approach, we argue, adds to the diversity of evaluation methods in the field and strengthens the paper by providing an alternative lens through which XAI methods can be assessed.

---

### Official Review · Reviewer_QfZD · 2023-10-30

**Soundness:** 3 good
**Presentation:** 3 good
**Contribution:** 2 fair
**Rating:** 3
**Confidence:** 5

**Summary:**

This paper introduces an axiomatic framework for generating concept-based explanations for a given model. This approach generalizes existing methods, such as TCAV and the completeness score, under specific conditions.

**Strengths:**

Interpretable and explainable machine learning holds substantial societal implications, warranting its significance as a crucial area of research.

Concept explanation emerges as a compelling avenue within the realm of interpretable machine learning.

The axiomatic approach to concept explanation is interesting and makes sense to me.

**Weaknesses:**

Some assertions within the paper are inaccurate. Specifically, the authors incorrectly categorize SHAP and LIME as model-specific techniques that necessitate white-box access to the model. Contrary to these claims, SHAP and LIME are model-agnostic and can operate on black-box models. They generate explanations by probing the model with varying input combinations to obtain SHAP and LIME scores.

Furthermore, the terminology used in the problem formulation section lacks precision. The term "measure" is ambiguously defined— it is unclear whether it refers to a "score" or "measurement," or to a "measure" in the context of measure theory. A more mathematically rigorous exposition of the problem setting would be beneficial.

While the abstract and the introduction draw a grand picture of a unified approach that encompasses different existing methods, but ultimately only discussed TCAV and completeness-aware explanations in the paper. The proposed measure is also somewhat similar to how [1] parameterizes a self-explaining neural network (e.g., the product of h(x) and c(x)).

Additional points of concern include: (1) the assertion that the proposed approach subsumes TCAV is not entirely accurate. According to Theorem 4.2, it approximates TCAV in an asymptotic sense rather than subsuming it exactly. (2) The claim that the approach unifies TCAV and completeness-aware explanations is also not entirely precise. Specifically, the class-conditioned variant of the proposed method is linearly correlated with a subset of the completeness score, while its concept-conditioned variant approximates the continuous case of TCAV.

From Section 3, it seems the proposed approach is limited to binary classification. This is a limitation. Would it be generalizable to other cases, e.g., multi-class classification?

Another significant issue that I have is the absence of comparative results with baseline methods. Unless I overlooked it, the paper does not present any performance metrics juxtaposed with at least TCAV and completeness-aware explanations. Such comparative analysis is essential for rigorously assessing the technical and empirical merits of the proposed approach. Without it, evaluating the method's substantive contribution becomes challenging.

[1] Towards Robust Interpretability with Self-Explaining Neural Networks. NeurIPS 2018.

**Questions:**

Would it be generalizable to other cases, e.g., multi-class classification?

The definition of “measure” in the problem formation paragraph is a bit vague. By “measure" do you mean a “score/measurement” or a “measure” as in the measure theory?

---

> ### Author Response · Authors · 2023-11-21
> **Thanks for your review**
>
> We would like to thank the review for their suggestions. Here are our response:
>
> **SHAP and LIME**. In our discussion of these methods in the related section, we highlight their model-agnostic nature, please refer to Section 2 paragraph “model-agnostic explanations”.
>
> **Problem formulation**. We formally defined the problem in section 3, stating that the objective is to find $M$, a function that maps the following parameters to $\mathbb{R}$: (i) the model $h$, (ii) the concept $c$, and (iii) the a priori probability $p(\cdot)$ over the examples.
>
> **“Assertion that the proposed approach subsumes TCAV…”** we aim to establish connections between our approach and TCAV, rather than make such assertion. We will revise our presentation to reflect this point more clearly.
>
> **Generalizable to multi-class classification.** Yes, it is possible to generalize to multi-class classification and select the relevant symmetry function in Axiom 3. For simplicity we focused on the basic setup of binary classification. Also see Section 3 “notation”, where we mentioned how it can be extended to multiclass by using the one-vs-all reformulation.
>
> **Comparative results.** Our goal in the paper is not to claim that our approach is better than TCAV or completeness-aware methods. This is why we chose not to compare them empirically. One exception to this rule is that we mentioned our method's faster computational speed compared to those approaches and provided an explanation for it  at the end of sections 4.1 and 4.2. However, we can also test it empirically. Thank you for your suggestion.

---

### Official Review · Reviewer_iv2S · 2023-10-31

**Soundness:** 2 fair
**Presentation:** 2 fair
**Contribution:** 2 fair
**Rating:** 3
**Confidence:** 4

**Summary:**

Their objective is to design a model-agnostic concept explainer; Existing concept explanation methods assume the model-to-be-explained has a representation layer limiting their application to DNNs.
They begin with a presentation of three axioms, which must hold for any concept explainer, and draw connections to previous estimators.
The paper claims that their framework enables efficient implementation and helps interpret concept scores estimated by existing estimators.

The paper does bring some structure to the problem but I have strong concerns regarding faithfulness of their overly-simplified estimator and reservations about their compute efficiency claims.

**Strengths:**

- The paper unifies and reinterprets two popular concept explanation techniques (Kim et.al. and Yeh et.al.), which is neat.
- The motivation for a model-agnostic explainer is a valid concern. Concept explanations are biased towards DNNs.

**Weaknesses:**

- **Presentation issues**. The notation is confusing. Concept is denoted with c(x) and is used to denote either set of concepts (i.e. a concept vector) or a scalar score interchangeably causing much confusion, see (1), (2). Yeh et.al. 2020 should be explained in more details in Sec 4.1. h is overloaded to map from example or representations to the label. (2) uses y to enumerate over concept values, which can be confusing.
- **Faithfulness is not established**. (This is less a weakness and more a show-stopper). The premise of an explanation is its faithfulness, which surprisingly is not one of the axioms. All their estimators at best measure the correlation in activation of concept and the model-to-be-explained. They need to establish the faithfulness of their estimator with the evaluation suite of TCAV or Yeh et.al. to first establish theirs as an explanation. Imagine a model trained on a dataset with two co-occurring concepts but the model may only exploit one of the concepts for prediction (say because one concept is way easier than the other), however none of their estimators can bring out the asymmetric concept importance.
- **Unconvincing evaluation**. The paper evaluates on a gamble of tasks. The prediction tasks considered are to predict *cat*, *chair*, *motorbike*, *bottle* in Figure 2, 3, 4 respectively. Some rationale on choice of tasks is expected, and some evaluation using established benchmarks is also expected.
- **Computational efficiency not presented.** They claim computational efficiency but do not present any empirical evidence supporting the claim.
- **Sufficiency vs necessary based score**. The paper discusses concept scores based on sufficiency or necessity, but do not discuss their practical implications. When is one better than the other?

**Questions:**

Please see above.

---
**Post-rebuttal comment**

I thank the authors for providing clarifications and for keeping it short.

A-Pascal may is a recognised benchmark in the CV community but it is not a standard benchmark for evaluating explanations. Besides, different sections of the paper considered widely different labels for evaluation or presentation without much explanation.

The concern regarding faithfulness with the paper is severe because their estimated importance scores seem model-agnostic. TCAV, for instance, accounts the effect of concept interventions on the representation to measure the importance. Nevertheless, experiments proving faithfulness are much needed.

---

> ### Author Response · Authors · 2023-11-21
> **Thanks for your review**
>
> We would like to thank the reviewer for their thoughtful comments. Please see our responses below.
>
> **Presentation issues**.Thank you for your suggestion regarding notation. We will incorporate it into future versions of the paper.
>
> **Faithfulness**. Thank you for providing an example to illustrate your point. It's important to note that ANY black-box explainer that only uses examples from the given distribution, which is the focus of our work, cannot detect the case of two different-impact co-occurring concepts.
>
> **Evaluation**. We used the a-Pascal dataset with its 20 classes, this dataset is acknowledged in the CV community. See [1,2,3]. The structure of this dataset also suits our task at hand.
>
> **Computational efficiency**. At the end of sections 4.1 and 4.2, we elaborate on why these methods are more efficient. Thank you for suggesting an empirical examination of this aspect as well.
>
> **Sufficiency vs necessary based score.** It's not a matter of one being better than the other. Rather, it's about understanding how a concept influences a prediction. We've explored two types of influence: necessary and sufficient. There isn't a definitive choice between them; the selection depends on your specific application.
>
> [1] Paco: Parts and attributes of common objects
>
> [2]: Ovarnet: Towards open-vocabulary object attribute recognition
>
> [3]: Open-vocabulary attribute detection

---

### Official Review · Reviewer_rtvs · 2023-11-09

**Soundness:** 2 fair
**Presentation:** 2 fair
**Contribution:** 2 fair
**Rating:** 3
**Confidence:** 3

**Summary:**

This paper addresses the problem of obtaining model-agnostic and human-understandable concept explanations of deep networks, specifically in the context of image classification. It first defines a set of general axioms that a concept explanation should follow, and shows that some of the existing approaches to concept explanations (e.g. TCAV) can be viewed as a part of this framework. It then defines measures that satisfy the axioms and explores their use for model selection (for models that use appropriate features), model optimizer selection (SGD vs AdamW), and prompt editing (to improve models).

**Strengths:**

1. The paper addresses the important problem of obtaining human-understandable faithful explanations of the decisions of deep networks, which is crucial for trusting model decisions.
1. It proposes a systematic, model-agnostic approach using a set of axioms that a concept explanation should satisfy. It shows that some of the prior work on concept explanations can be placed within this general framework.

**Weaknesses:**

1. The experimental evaluation appears to be limited. The analysis in Section 5.2 is limited to a small set of classes, and it is unclear how generally the trends observed hold. The evaluation for model selection also uses very simple models (linear regression and random forest). For broader applicability, it would be useful to evaluate whether this generalizes to deep networks, and whether one can select between different networks and network architectures with this approach.
1. The evaluations in Sectionn 5.2 appear to focus on images that were predicted to be a given class. For different models and different optimizers, this set of images could be different. Given that the ground truth labels of the images do not seem to have been taken into account, it is unclear if the presence of "spurious" concepts observed is due to the model relying on spurious correlations for images where the ground truth class is the same as the predicted class being considered, or due to the model simply misclassifying images of other classes as the predicted class being considered. This would be important to properly evaluate the proposed approach.
1. Given the efficiency and equivalence arguments between the proposed approach and prior work (completeness-aware and TCAV, Section 4), it would be helpful to have an experimental comparison between these approaches in the application settings in Section 5, both in terms of performance and computational efficiency.
1. The formulation of TCAV as an instantiation of the proposed framework seems to only apply to explanations at the last layer, while the TCAV formulation as originally proposed is more general. If this is the case, then this should be made clear.

**Questions:**

Clarifications on the points raised in Weaknesses would be helpful.

---

> ### Author Response · Authors · 2023-11-21
> **Thanks for your review**
>
> We would like to thank the reviewer for their thoughtful comments. Please see our responses below.
> 1. In Section 5.2.1, we deliberately utilized linear regression and random forest to demonstrate one of the benefits of our method. Specifically, previous methods were tailored for neural networks, whereas our approach also accommodates other methods like linear regression and random forest models. Additionally, we also employed neural networks, detailed in sections 5.2.2 and 5.2.3.
> 2. Spurious correlation refers to the scenario wherein a model learns to depend on features that lack a causal relationship with the target variable but exhibit strong correlation within the training data. Our measure identifies concepts on which a model relies, enabling human judgment regarding their relevance to accurate predictions—thereby detecting spurious correlations. While dissecting the reasoning behind spurious correlation, as you suggested, could be interesting, it extends beyond the scope of the current research.
> 3. At the end of sections 4.1 and 4.2, we elaborate on why these methods are more efficient. Thank you for suggesting an empirical examination of this aspect as well.
> 4. We apologize for any confusion caused. We will make sure to provide clarification in future versions of the paper.

---

### Meta-Review · Area_Chair_5FFT · 2023-12-15

**Metareview:**

The paper proposes axioms to which a concept-based explanation should conform. All reviewers see the relevance of the direction and the value of the proposed axioms, while they also find many limitations with regards to the experiments and comparison to prior works, especially TCAV, that renders empirical conclusions difficult. The paper has merits and should become stronger in the next submission if it follows the reviewers' feedback for improved empirical evidence. The AC recommends rejection for the current submission.

**Justification For Why Not Higher Score:**

The paper is not ready for publication mainly due to the conclusivity of the experiments, comparisons, and side studies.

**Justification For Why Not Lower Score:**

N/A

---

### Decision · Program_Chairs · 2024-01-16

Reject